# Preference-Driven Classification Measure

**DOI:** 10.3390/e24040531

**Published:** 2022-04-10

**Authors:** Jan Kozak, Barbara Probierz, Krzysztof Kania, Przemysław Juszczuk

**Affiliations:** 1Department of Machine Learning, University of Economics in Katowice, 1 Maja 50, 40-287 Katowice, Poland; barbara.probierz@ue.katowice.pl (B.P.); przemyslaw.juszczuk@ue.katowice.pl (P.J.); 2Department of Knowledge Engineering, University of Economics in Katowice, 1 Maja 50, 40-287 Katowice, Poland; krzysztof.kania@ue.katowice.pl

**Keywords:** classification measure, quality of classification, quality measure, preference-driven classification, machine learning

## Abstract

Classification is one of the main problems of machine learning, and assessing the quality of classification is one of the most topical tasks, all the more difficult as it depends on many factors. Many different measures have been proposed to assess the quality of the classification, often depending on the application of a specific classifier. However, in most cases, these measures are focused on binary classification, and for the problem of many decision classes, they are significantly simplified. Due to the increasing scope of classification applications, there is a growing need to select a classifier appropriate to the situation, including more complex data sets with multiple decision classes. This paper aims to propose a new measure of classifier quality assessment (called the preference-driven measure, abbreviated p-d), regardless of the number of classes, with the possibility of establishing the relative importance of each class. Furthermore, we propose a solution in which the classifier’s assessment can be adapted to the analyzed problem using a vector of preferences. To visualize the operation of the proposed measure, we present it first on an example involving two decision classes and then test its operation on real, multi-class data sets. Additionally, in this case, we demonstrate how to adjust the assessment to the user’s preferences. The results obtained allow us to confirm that the use of a preference-driven measure indicates that other classifiers are better to use according to preferences, particularly as opposed to the classical measures of classification quality assessment.

## 1. Introduction

Classification continues to be one of the most important subjects in machine learning. Despite this, we still lack a general measure of quality independent of the specific characteristics of the data set. Moreover, in the situations where there is a necessity to involve the human decision maker in the classification process, we are forced to switch between different measures. Among the general ones, there are accuracy, precision or recall, and others that are data-dependent. Choosing the right (optimal) one is especially important because choosing a particular classification method depends heavily on the calculated quality measures.

Moreover, there is no single best classification measure that effectively identifies the method suitable for every task. Classification algorithms/methods have many characteristics. Consequently, there are many measures of classification because there is no single measure covering all the characteristics simultaneously [1]. Thus, finding an appropriate classification measure for a specific task is difficult and requires answering the question of what conditions, in specific circumstances, must be met by the measure.

One of the ways to bypass the problem of unambiguous assessment of the classifier’s quality and selecting one best-suited classifier is ensemble and hybrid methods, which simultaneously use many different algorithms to perform a specific task. Within this approach, we can point out the homogeneous and heterogeneous solutions [2]. In the first group, we can find methods allowing us to create large groups of classifiers belonging to the same category (or even classifiers generated with the same method, but with different starting parameters), which allow the classification process simultaneously. The heterogeneous approach involves using a large variety of classifiers, for which the main advantage is the diversity of obtained results. Thus, the basic idea of these methods is the idea of collective decision making [3].

One should note that the above approach based on the ensemble methods allows for more robust classifier selection. However, it still leaves the decision maker with the problem of estimating the classification quality. In medicine, military or finance, the well-known accuracy measure seems to derive unsatisfactory results and present limited usability [4,5]. On the other hand, measures such as recall or precision are directed towards the binary classification problem. Most of the proposed measures are directly connected to the confusion matrix and related absolutely to the numerical outcome of the classification [6]. In most cases, it is strongly needed, but it makes the whole process independent of the user’s preferences. In the decision-making context, taking them into account may be vital to make the process effective and at the same time maintain the sovereignty of the decision maker.

Incorporating users in the process of preparing a machine learning solution is an essential element of the entire procedure, the subject of many studies, and can take various forms [7,8]. One of the goals of the actions taken is to help the user to choose a suitable classifier. Most often, this task comes down to comparing simple measures of classification quality, which is usually carried out by trial and error, and yet can be unreliable [9]. This task becomes even more complicated if individual users’ preferences are to be taken into account. This applies especially to issues of a managerial nature, but more generally wherever a human being to some extent participates in the decision-making process. In practice, this applies to all issues except physical or technical ones, where only objective laws are in force [10]. The main intention of introducing the new measure is to make this stage of a research procedure more methodical. To the best of our knowledge, there are no clear guidelines for taking into account the parameters in the learning process and the selection of a classifier in conjunction with individual preferences. The next issue is the systematic classification into the following application areas, thus expanding the group using machine learning methods. Finally, some users need a tool to control the process of selecting a classifier for their own needs, which are more complex and related to many classes. For such users, a measure that allows them to simply, directly, and methodically include their preferences in selecting and training the classifier would be very beneficial.

To cope with the above drawbacks, and at the same time maintain the role of the decision maker in the process, we propose the idea of a new measure in which their preferences are vital to the importance of individual classes. We aim to propose a measure that balances a thorough analysis of the classifier’s performance and the selection of the classifier that performs best under the conditions (preferences) specified by the user.

The main intention of introducing the new measure is to make this stage of the research procedure more methodical. Users need a tool with which they will be able to select a classifier for their own needs, which are more complex and related to many classes. A measure that allows them to directly and methodically include their preferences in selecting and training the classifier would be very beneficial for such users.

We undertook to propose such a measure because, to the best of our knowledge, there is no reasonable alternative where, for any number of decision classes, it is possible to aggregate the quality of classification depending on the weight for a particular decision class. First, our solution was discussed in detail on prepared examples for two decision classes. It was then tested on real-world data sets and re-examined for a more significant number of decision classes that occurred in these data sets.

This article is organized as follows. Section 1 introduces the subject of this article. Section 2 provides an overview of the work related to the classification, particularly the measures for assessing the quality of the classification. In Section 3, we describe the classification problem and the classification quality assessment measures based on the error matrix for binary and multi-class classification. In Section 4, we present a new measure for classification, in which it will be possible to control preferences. In Section 5, we present the analysis of our research on real data sets. Finally, in Section 6 and Section 7, we discuss the results of the experiments and end with general remarks on this work and available directions for future research.

## 2. Related Works

Evaluating the classification performance is a difficult task, and the discussion on this topic arose from the beginning of work on automatic classification. The initial set of five measures (sensitivity, specificity, efficiency, positive and negative predictive value) was rapidly expanded [11]. The most often used measure of classification performance is accuracy. However, it is not the only measure of the quality of predictive models. Despite optimizing the classification error rate, high-accuracy models may fail to capture crucial information transfer in the classification task [12,13]. Despite the simplicity and intuitive interpretation, there are many reasons and situations in which accuracy should not be used [14]. Instead, the authors advocate for using Cohen’s kappa as a better meter for measuring classifiers’ own merits than accuracy. Moreover, [15] indicates that the most frequently used measures, which focus on correctly classified cases (precision, recall, or F-score), do not meet the needs of various decision-making situations, especially when more than one class is essential. The authors advocate for using three other measures—Youden’s index, likelihood, and discriminant power—because they combine sensitivity and specificity and their complements.

While most studies concern binary classification, in [16], the authors focused on multi-class classification problems. The authors showed that the extension of measures to the classification of many classes is associated with averaging the results achieved for individual classes in most cases. Finally, in [17], the authors point out some shortcomings of the accuracy measure and list five conditions that the newly constructed discriminator metric should meet.

To solve the dilemma related to the choice of a measure for a given problem, a list of desired features of an ideal measure and analysis of the most known measures was proposed in [1]. More importantly, they proved that it is impossible to satisfy them simultaneously. They also proposed a new family of measures (Generalized Means) that meet all desirable properties except one, and a new measure called Symmetric Balanced Accuracy.

A comparative analysis and taxonomy of the quality of classification measures have been the subject of many studies. For instance [18], in their experimental comparison conducted on 30 data sets, proposed dividing the performance measures into three categories:Measures based on a threshold, such as accuracy, modified accuracy measure, F-measure, or Kappa statistic, which are used to minimize the number of errors in the model. They are based directly on a confusion matrix, and they are widely used in many classification tasks. One should note that the overall efficiency of these measures is strictly related to the quality of the data. However, some measures, such as accuracy, can be less effective in the case of unbalanced data sets.Measures based on a probabilistic approach to understanding error, i.e., measuring the deviation or entropy information between the actual and predicted probability, such as mean absolute error, mean square error (Brier score), LogLoss (cross-entropy). These measures are useful in measuring the probability of selecting the wrong class, which is essential in ensemble methods or for a committee of classifiers.Measures based on the model’s ability to correctly rank cases include ROC, AUC, the Youden index, precision–recall curve, Kolmogorov–Smirnov, or lift chart. They are helpful when indicating the best *n* occurrences in a data set or when good class separation is needed. They are widely used in recommendation systems, design marketing campaigns, fraud detection, spam filtering, and more.

In a survey [19], the authors grouped the measures depending on the type of outcome on which a given measure is focused (correct or incorrect outcome). In turn, [20] presents an in-depth analysis of over twenty performance measures used in different classification tasks in the context of changes made to a confusion matrix and their relations with particular measures (measure invariance). In contrast, a comparative study of two or more classifiers based on statistical tests was presented in [21]. A comprehensive analysis of the methods and measures of classification assessment is also included in [22], where the relationships between all measures calculated based on the confusion matrix are shown. Finally, a different approach to the analysis of performance measures was presented in [23]. First, the authors grouped classification measures according to classification difficulty, which they defined in relation to a distance between the boundary lines and each correctly classified case. The authors later developed their idea and proposed an instance-based measure for calculating the performance of classification from the perspective of instances, called degree of credibility [24].

The set of measures is constantly growing. For instance, in [25] was proposed a measure that compares classifiers, which combines three measures from different groups: Matthews correlation coefficient as a measure, which is calculated from both true and false positives and negatives, and AUC (area under the curve), derived from ROC and accuracy. To overcome the shortcomings of the accuracy measure in evaluating multi-class classifiers and to improve the quality of classifiers, in [26], the authors proposed a metric based on the combined accuracy and dispersion values. They also showed experimentally that this two-dimensional metric is particularly suitable in complex, unbalanced data sets and with many classes.

The new measures are also proposed to supplement the already used measures that work better for specific tasks. For example, [27], as an alternative to measures used in medical diagnostics, which use only part of the values from the error matrix, define the measure AQM, which takes into account all values from the confusion matrix. Similarly, in [28], for image analysis, a new measure of classification performance called robust-and-balanced accuracy was introduced. It aims to connect balanced accuracy with measures of variations. In another proposition, to improve face recognition processes, a new classification measure, called the volume measure, based on the volume of the matrix, was proposed [29]. In turn, a measure dedicated to the analysis of imbalanced data sets based on the harmonic mean of recall and selectivity was proposed in [30].

Existing measures are also modified. For example, in [31], the $F^{*}$ measure was proposed as a modification of the F-score, towards the more straightforward interpretation of this measure. The above short review shows that the issue of classification measures is constantly under the attention of researchers. Moreover, for specific applications, new measures are created that are better suited to the requirements of the domain or user preferences.

## 3. Classification and Quality Evaluation

Formally, the classification problem *Q* can be solved using empirical experience *W*, while the quality of the solution is estimated by the quality measure *Y*. The value of *Y* should be increasing, while the experience *W* rises as well [32].

In machine learning, classification refers to the prediction problem of determining the class to which samples from a data set will be assigned. A classifier algorithm (often shortened to “a classifier”) must be provided with training data with labeled classes. Then, the classifier can predict classes for new test data based on the training data. This approach is called supervised machine learning, and classification is one example of such a method. The training set is selected as a subset of the whole data set. A typical approach is to divide the known examples into a train and test set, following some general principles about the ratio of the two. Eventually, the test set includes a far smaller number of samples than the training set (preferably, the test set and training set should be disjoint). Test set is used to evaluate classification quality. Every object (also called a sample or observation) from the training data is assigned some predefined label (decision class). The idea is to build such a classifier, which assigns the proper labels for the objects. In contrast, the evaluation is performed on the training set, where the difference between the assigned and the actual label is estimated.

Classification algorithms for a prediction problem are evaluated based on performance. Multiple measures for assessing classification quality can be used depending on the situation. One of the most commonly used measures is accuracy, which determines how many samples from the entire data set were correctly classified into the appropriate classes. Often, other measures are used in addition to accuracy that more accurately assess the classifier’s performance. Such measures primarily include precision and recall. Unfortunately, there are many problems in which classical measures are insufficient, and it is necessary to look for new solutions.

Let us consider a set of all available samples (called the universe of objects) *X*, which will include *n* number of objects: (1)$$X=\{x_{1},x_{2},\cdots ,x_{i},\cdots ,x_{n}\},$$

A single observation $x_{i}$ will be described by a finite set of attributes and the decision attribute: (2)$$a_{1},a_{2},\cdots ,a_{m},$$
where $a_{j}\in A_{j}$, j=1,⋯,m. In this context, $A_{j}$ denotes the domain of the *j*-th attribute, while features $a_{1},a_{2},\cdots ,a_{m}$ create a feature space $A_{1}\times A_{2}\times \ldots \times A_{m}$.

There are no general restrictions related to the values of attributes, which can be quantitative or categorical. However, one should note that preprocessing allows for discretizing selected quantitative attributes. The same procedure can be applied to the decision classes, where many labels can be merged into fewer during the discretization process (trade-off between the quality of classification and classification speed). Thus, the single sample can be described as follows: (3)$$x_{i}=\left(\vec{V_{i}},c_{k}\right),v_{i}^{j}\in A_{j},c_{k}\in \left\{1,\cdots ,C\right\},$$
where $\vec{V_{i}}=\left[v_{i}^{1},\cdots ,v_{i}^{m}\right]$ is a vector in an *m*-dimensional feature space, $v_{i}^{j}$ is the value of attribute $a_{j}$ for observation (sample) $x_{i}$, and $c_{i}$ is the class label (also called the decision class) of this object. Thus, the universe *X* can be formally described as: (4)$$X:{\left\{\left(\vec{V_{i}},c_{i}\right)\right\}}_{i=1}^{n}.$$

Hence, the classification problem can generally be understood as the assignment problem, where every element $x_{i}$ from the universe *X* should have the decision class $c_{i}$ assigned. Eventually, we end with the classifier capable of assigning the newly arrived objects from the test set into proper decision classes. However, even in the case of the binary classification problem, the above is not trivial. It can be challenging to achieve when, for example, we observe unbalanced data (i.e., the situation in which most of the objects from universe *X* are assigned to a single class). Therefore, many different estimation methods were proposed (both to binary and multiple-class classification problems) to cope with this. Next, this section discusses various classification measures capable of dealing with both mentioned classification problems.

### 3.1. Quality Assessment and Binary Confusion Matrix

A confusion matrix is among the most popular tools used in the process of the validation of the quality of performed classification. The confusion matrix for a binary decision class is defined as a table with different combinations of predicted and actual values related to the decision classes [33,34]. The rows contain existing classes, while the columns contain predicted classes. The diagonal of the confusion matrix includes the correctly classified samples for both classes, while the off-diagonal cells represent the errors (misclassified samples for both decision classes). The confusion matrix represents the errors that the classifier makes and shows the type of these errors. It represents a detailed breakdown of the answers considering the number of correct and incorrect classifications for both classes. It is imperative when the cost of misclassification is different for these classes and when the size of the classes is different [35].

For the binary classification (e.g., classification of emails and spam or sick and healthy patients), see Table 1, where the prediction of a positive class (labeled 1) and a negative class (labeled 2) can be uniquely determined. Such a situation indicates whether the classifier is more likely to incur an error by assigning a positive class as a negative class or vice versa. In Table 1, the confusion matrix for the binary classifier is shown, where:TP (true positive)–samples classified as predicted class 1, which are samples of actual class 1;FN (false negative)–samples classified as predicted class 2, which are samples of actual class 1;FP (false positive)–samples classified as predicted class 1, which are samples of actual class 2;TN (true negative)–samples classified as predicted class 2, which are samples of actual class 2.

The quality of classifiers is measured based on a confusion matrix (as shown in Table 1). Measures based on the confusion matrix include, but are not limited to, accuracy, precision, recall, and F-score (also called F1, which is the measure $F_{\beta }$ for which β=1). In addition, the Matthews correlation coefficient (MCC) measure and BalancedAccuracy are also often used for binary classification.

The accuracy measure indicates how often a classifier makes a correct prediction. It is the ratio of the number of accurate predictions to the total number of predictions and is calculated based on Formula (Equation 5).
(5)$$accuracy\_binary=\frac{TP+TN}{TP+FP+FN+TN}$$

Precision determines how many samples, out of all those classified as positive, are samples of the positive class. Precision is expressed by Formula (Equation 6).
(6)$$precision\_binary=\frac{TP}{TP+FP}$$

Recall is used to determine how many samples belonging to a positive class were classified as positive by the classifier. Recall is determined by Formula (Equation 7).
(7)$$recall\_binary=\frac{TP}{TP+FN}$$

For binary classification, the measure $F_{\beta }$ is defined as the harmonic mean of precision and recall, where additional precision or recall weights are used to obtain more accurate results. By setting the value of β, it is possible to control the effect of recall weight with respect to precision. $F_{\beta }$ is specified by Formula (Equation 8), where β is the number of times the recall is as important as the precision [36]. The $F_{\beta }$ value ranges from 0 to 1 (with 0 being the worst value and 1 being the optimal value). The most common value for β is 1, which simply means measure F1 (Equation (Equation 9)). Other frequently used values are 2 and 0.5. In the case of 2, recall weight is greater than precision; however, in the case of 0.5, recall weight is less than precision. The $F_{\beta }$ measure is based on the Van Rijsbergen measure of effectiveness [37].
(8)$$F_{\beta }=\left(1+\beta^{2}\right)\times \frac{precision\_binary\times recall\_binary}{\beta^{2}\times precision\_binary+recall\_binary}$$

The F1 measure is the instance of the measure $F_{\beta }$ (Equation (Equation 8)), for which β=1. The F1 measure is defined as the harmonic mean of precision and recall [38]. Therefore, F1 will only take on a high value if both of its components reach a high value. The F1 measure often replaces precision when class counts are unbalanced [39]. For example, if 97% of the data belong to class 1 and only 3% belong to class 2, then classifying all observations as class 1 would yield a misleading accuracy of 97%. The F1 measure is based on precision and recall, and is thus robust to such distortions. The measure is calculated from Formula (Equation 9).
(9)$$F1\_binary=2\times \frac{precision\_binary\times recall\_binary}{precision\_binary+recall\_binary}=\frac{TP}{TP+\frac{1}{2}\left(FN+FP\right)}$$

The Matthews correlation coefficient (MCC) measure is often used for unbalanced data [40]. For the precision, recall, and F1 measures, the TN value is not used, which is very important if we are interested in both classes. Therefore, the Matthews correlation coefficient, calculated based on all terms from the confusion matrix, can be used. The MCC measure is calculated from Formula (Equation 10).
(10)$$MCC\_binary=\frac{TP\times TN-FP\times FN}{\sqrt{\left(TP+FP)(TP+FN)(TN+FP)(TN+FN\right)}}$$

The MCC value ranges within [−1…1] (with −1 equal to the misclassification of all samples, while a value of 1 means that all samples are correctly classified). Therefore, the higher the correlation between actual and predicted values, the better the prediction. Furthermore, for the MCC measure, the two classes have the same importance weight, so when positive classes are swapped with negative classes, the MCC value will be the same, which means that the MCC measure is symmetric [40].

Another measure in binary classification is the BalancedAccuracy measure, which calculates balanced accuracy for anomaly or disease detection, where significant differences in the class size are observed [41]. Overestimated accuracy results can be avoided using the BalancedAccuracy measure for unbalanced classes. For binary classification, the balanced accuracy equals the arithmetic mean of the recall and specificity. The BalancedAccuracy measure is calculated by Formulas (Equation 11) or (Equation 12).
(11)$$BalancedAccuracy\_binary=\frac{recall\_binary+specificity\_binary}{2}$$
(12)$$BalancedAccuracy\_binary=\frac{1}{2}\left(\frac{TP}{TP+FN}+\frac{TN}{FP+TN}\right)$$

Specificity is used to determine the number of samples belonging to a negative class that have been classified as negative by the classifier [42]. Specificity is measured by Formula (Equation 13).
(13)$$specificity\_binary=\frac{TN}{FP+TN}$$

### 3.2. Quality Assessment and Confusion Matrix for Multi-Class Classification

In addition to binary classification, it is common to classify samples into more than two classes. We are dealing with multi-class classification in such a situation—not to be confused with multi-label classification. The difference between multi-class and multi-label classification is that a sample can only be assigned to one class selected from multiple classes for multi-class classification. In contrast, a sample can be assigned to multiple classes for multi-label classification [43].

There is a need to construct an extended confusion matrix for a number of decision classes greater than 2. Thus, the number of rows and the number of columns in such an approach will equal the number of classes. For this purpose, we present the idea of a confusion matrix for *C* classes, which is shown in Table 2. Similarly, for the binary classification problem (see Table 1), we have listed four terms:TP (true positive) means that samples from the actual class have been classified into the same predicted class—denoted as $TP_{1},\;TP_{2},\ldots ,\;TP_{i},\ldots ,\;TP_{C-1},\;TP_{C}$;FN (false negative) indicates that samples in the actual class have been classified into other predicted classes. This is the sum of the values of the corresponding row of the real class except for the TP values for that real class—denoted as $FN_{1},\;FN_{2},\ldots ,\;FN_{i},\ldots ,\;FN_{C-1},\;FN_{C}$;FP (false positive) indicates that samples from other real classes have been classified into the selected predicted class. This is the sum of the values of the corresponding column of the predicted class except for the TP values for the actual class—denoted as $FP_{1},\;FP_{2},\ldots ,\;FP_{i},\ldots ,\;FP_{C-1},\;FP_{C}$;TN (true negative) indicates that, for the selected real class, samples from other actual classes were classified into predicted classes other than the predicted class corresponding to the chosen real class. For a given real class, it is the sum of the values of all columns and rows except for the row and column values of that real class for which we compute the values—denoted as $TN\backslash_{\left\{i\right\}}$ or $TN\backslash_{\left\{i,C\right\}}$.

To evaluate the quality of multi-class classification, as in binary classification, measures based on the confusion matrix shown in Table 2 were used. Therefore, referring to the measures described in Section 3.1, we wish to present them for the multi-class classification problem.

Accuracy is one of the most commonly used measures in a multi-class classification problem [16] and is calculated according to Formula (Equation 14) or (Equation 15); when we define the sum of all samples as *s*, this equation boils down to the form (Equation 16). To calculate accuracy, sum all correctly classified samples and then divide by the number of all classified samples. Correctly classified samples are shown in the confusion matrix (see Table 2) on the main diagonal (from upper left corner to lower right corner).
(14)$$accuracy=\frac{\sum_{i=1}^{c}TP_{i}}{\sum_{i=1}^{c}TP_{i}+\sum_{i=1}^{c}FP_{i}}$$
(15)$$accuracy=\frac{\sum_{i=1}^{c}TP_{i}}{\sum_{i=1}^{c}TP_{i}+\sum_{i=1}^{c}FN_{i}}$$
(16)$$accuracy=\frac{\sum_{i=1}^{c}TP_{i}}{s}$$

Precision and recall are used for the multi-class approach as well. Two modifications can be distinguished in this case: macro− and micro− precision or recall [16]. For the macro− modification, to calculate the value of the measures for multiple classes, one must count precision and recall for each class separately and then calculate the arithmetic mean of these values. In this way, all classes during multi-class classification have the same validity, regardless of the class count. In multi-class classification, macro_precision is calculated by Formula (Equation 17) and macro_recall is calculated by Formula (Equation 18).
(17)$$macro\_precision=\frac{1}{c}\sum_{i=1}^{c}\frac{TP_{i}}{TP_{i}+FP_{i}}$$
(18)$$macro\_recall=\frac{1}{c}\sum_{i=1}^{c}\frac{TP_{i}}{TP_{i}+FN_{i}}$$

In contrast, for micro_ modification, to count the precision and recall values, one must look at all classes together. In this way, each correctly classified sample into a class is a component of all correctly classified samples. In other words, we calculate TP as the sum of all TP values for individual classes (the sum of the values from the main diagonal). The FP value will be the sum of all values off the main diagonal, equal to the FN value. Therefore, micro_precision and micro_recall are the same because they are the sum of TP values to all values in the confusion matrix [16]. In multi-class classification, micro_precision is calculated by Formula (Equation 19) and micro_recall is calculated by Formula (Equation 20).
(19)$$micro\_precision=\frac{\sum_{i=1}^{c}TP_{i}}{\sum_{i=1}^{c}TP_{i}+\sum_{i=1}^{c}FP_{i}}$$
(20)$$micro\_recall=\frac{\sum_{i=1}^{c}TP_{i}}{\sum_{i=1}^{c}TP_{i}+\sum_{i=1}^{c}FN_{i}}$$

The $F_{\beta }$ measure is also used for multi-class classification problems; however, the results of this measure are obtained by macro-averaging or micro-averaging [20]. When we assume that all classes are of the same weight, we use macro-averaging, where two additional methods can be adopted. The first is the calculation of the $F_{\beta }$ measure from the macro_precision (see Equation (Equation 17)) and macro_recall (see Equation (Equation 18)) measures, and the second is the arithmetic mean of the F-beta scores for each class separately, based on the $F_{\beta }$ measure for binary classification (see Equation (Equation 8)). The $macro\_F_{\beta }$ measure is specified by Formula (Equation 21).
(21)$$macro\_F_{\beta }=\left(1+\beta^{2}\right)\times \frac{macro\_precision\times macro\_recall}{\beta^{2}\times macro\_precision+macro\_recall}$$

In contrast, in the classification depending on the frequency of classes, the $F_{\beta }$ results are calculated by micro-averaging based on micro_precision (see Equation (Equation 19)) and micro_recall (see Equation (Equation 20)). The $micro\_F_{\beta }$ measure is specified by Formula (Equation 22). In all of the discussed variants of the $F_{\beta }$ measure, the beta parameter determines the weight of the reference in relation to the precision. When β<1, precision has more weight, and when β>1, recall has more weight [20].
(22)$$micro\_F_{\beta }=\left(1+\beta^{2}\right)\times \frac{micro\_precision\times micro\_recall}{\beta^{2}\times micro\_precision+micro\_recall}$$

The F1 measure (the most common instance of $F_{\beta }$, where β=1) is calculated based on recall and precision (present in the multi-class classification) [44], and thus can be used in the multi-class classification as well. For a problem where class imbalance does not matter and all classes are equally valid, we use the macro_ modification and apply macro_precision (see Equation (Equation 17)) and macro_recall (see Equation (Equation 18)). Thus, the measure macro_F1 can be calculated according to Formula (Equation 23).
(23)$$macro\_F1=2\times \frac{macro\_precision\times macro\_recall}{macro\_precision+macro\_recall}$$

In problems with unbalanced classes, where selected classes are more important than others, the micro_ modification leads to the micro_precision (see Equation (Equation 19)) and micro_recall (see Equation (Equation 20)). Thus, the measure micro_F1 can be calculated according to Formula (Equation 24).
(24)$$micro\_F1=2\times \frac{micro\_precision\times micro\_recall}{micro\_precision+micro\_recall}$$
From Formulas (Equation 19), (Equation 20) and (Equation 24), one can conclude that, for the multi-class classification problem, the values of the measures micro_precision, micro_recall, and micro_F1 are equal to each other. At the same time, the values of these measures are equal to the accuracy of Formula (Equation 16).

The Matthews correlation coefficient (MCC) is only used in classifications of up to two classes (see Equation (Equation 10)). For classifications with more than two classes, it is often irrelevant to determine the division of multiple classes into two classes (positive and negative) [45]. Therefore, J. Gorodkin, in his work [46], proposed an extended correlation coefficient (called the $R_{K}$ statistic, for *K* different classes) that can be used in multi-class classification. Based on this, we defined MCC for multiple classes denoted as Formula (Equation 25), where *s* is the sum of all samples, $TP_{i}+FP_{i}$ is the value of all samples in row *i*, and $TP_{i}+FN_{i}$ is the value of all samples in column *i*.
(25)$$MCC\_multiclass=\frac{s\times \sum_{i=1}^{c}TP_{i}-\sum_{i=1}^{c}\left(\left(TP_{i}+FP_{i}\right)\times \left(TP_{i}+FN_{i}\right)\right)}{\sqrt{s^{2}-\sum_{i=1}^{c}{\left(TP_{i}+FP_{i}\right)}^{2}}\times \sqrt{s^{2}-\sum_{i=1}^{c}{\left(TP_{i}+FN_{i}\right)}^{2}}}$$

The last discussed measure is BalancedAccuracy, which could be used to calculate accuracy for unbalanced classes [41]. According to Formula (Equation 11), BalancedAccuracy_binary is the arithmetic mean of recall_binary and specificity_binary. For binary classification, the value of specificity_binary for the first class equals recall_binary for the second class. For this reason, in multi-class classification, to calculate BalancedAccuracy, one must count the recall for each class separately and then calculate the arithmetic mean of these values, according to Formula (Equation 26).
(26)$$BalancedAccuracy=\frac{1}{c}\sum_{i=1}^{c}\frac{TP_{i}}{TP_{i}+FN_{i}}$$

## 4. Preference-Driven Quality Assessment for Multi-Class Classification

Considering the measures described above, the preferences for individual classes in the classifier quality assessment are insufficient because there is no place in their construction to indicate such preferences. Thus, we propose a new preference-driven classification quality evaluation measure to evaluate classification quality based on different weights for each decision class. The proposed measure works independently of the number of decision classes—its definition is the same in binary and multi-class classification. We also suggest default values for this measure that can be used, in the test case, without specifying exact preferences for each decision class.

### 4.1. Proposed Preference-Driven Classification Measure

According to the measures given in Section 3.2, the proposed measure was defined based on the confusion matrix given in Table 2. We aim to keep it as simple as possible while satisfying the assumption of adjustment to preferences (of each decision class). Therefore, the proposed measure is based on a confusion matrix and precision and recall measures with κ parameters determining their relative importance.

The preference-driven classification measure is denoted as $\mathrm{preference}-{\mathrm{driven}}_{\vec{\kappa }}$, where $\vec{\kappa }$ is the preference vector, whose length is equal to the number of decision classes (see Formula (Equation 27)). The κ weights for each of the subsequent measures are written on the subsequent positions of the vector. The higher the κ value for a given decision class, the greater the importance of precision (determined by Formula (Equation 28)) relative to recall (determined by Formula (Equation 29))—based on this class only. Therefore, changing the κ values of a given class makes it possible to control the relative importance of precision and recall. This is a multi-criteria process because the κ value can differ for each class.

Finally, the $\mathrm{preference}-{\mathrm{driven}}_{\vec{\kappa }}$ measure can be expressed by Formula (Equation 30). One should note that $\vec{\kappa }$ is a parameter related to the measure by which the relative importance between precision and recall can be established for each decision class separately. For example, $\vec{\kappa }=\left[0.2,0.6,0.3\right]$ means that, for the first class, 20% precision and 80% recall are used; for the second class, 60% precision and 40% recall are used, and for the third class, 30% precision and 70% recall are used. To keep the final value of the $\mathrm{preference}-{\mathrm{driven}}_{\vec{\kappa }}$ measure in the range [0.0, 1.0], the sum of all these values is divided by the number of classes.
(27)$$preference-driven_{\vec{\kappa }}=\frac{1}{c}\sum_{i=1}^{c}\kappa_{i}\times \frac{TP_{i}}{TP_{i}+FP_{i}}+\left(1-\kappa_{i}\right)\times \frac{TP_{i}}{TP_{i}+FN_{i}}$$
(28)$$precision_{i}=\frac{TP_{i}}{TP_{i}+FP_{i}}$$
(29)$$recall_{i}=\frac{TP_{i}}{TP_{i}+FN_{i}}$$
(30)$$preference-driven_{\vec{\kappa }}=\frac{1}{c}\sum_{i=1}^{c}\kappa_{i}\times precision_{i}+\left(1-\kappa_{i}\right)\times recall_{i}$$

### 4.2. Proposed Measure Analysis in the Test Case

The sample confusion matrix was prepared for a classification problem with two decision classes to demonstrate how the measure works depending on the preference and classification outcome. The two classes were chosen to visualize the measure’s values.

Figure 1 presents nine confusion matrices for which $\mathrm{preference}-{\mathrm{driven}}_{\vec{\kappa }}$ values were determined for $\vec{\kappa }$, being a combination of all values from 0.0 to 1.0 with a step of 0.1, i.e.,
[0.0,0.0],[0.0,0.1],⋯[0.5,0.4],[0.5,0.5],[0.5,0.6],⋯[1.0,0.9],[1.0,1.0].
It gives a total of 121 different combinations vectors of preferences. The results are visualized in the following figures.

To better capture the distribution of values for the preference-driven measure, an analysis based on the confusion matrix cm4 (see Figure 1) was presented. It was selected because the quality assessment in terms of classical measures always means the same values, i.e., precision (see Equation (Equation 17)) is 0.7500, and recall (see Equation (Equation 18)) is 0.8333. In Figure 2, one can see that the value of the preference-driven measure ranges from 0.5833 for $\vec{\kappa }=\left[1.0,0.0\right]$ to 1.0000 for $\vec{\kappa }=\left[0.0,1.0\right]$. It is possible to obtain exactly the same values for recall ($\vec{\kappa }=\left[0.0,0.0\right]$) and precision ($\vec{\kappa }=\left[1.0,1.0\right]$), but depending on the preference, the classifier will be evaluated differently. For example, $\mathrm{preference}-{\mathrm{driven}}_{\left[0.1,0.3\right]}$, i.e., for $\kappa_{1}=0.1$ and $\kappa_{2}=0.3$ is 0.8083 (green dot in Figure 2); similarly, $\mathrm{preference}-{\mathrm{driven}}_{\left[0.1,0.8\right]}=0.9333$ (black dot in Figure 2), $\mathrm{preference}-{\mathrm{driven}}_{\left[0.4,0.4\right]}=0.7833$ (red dot in Figure 2), $\mathrm{preference}-{\mathrm{driven}}_{\left[0.9,0.1\right]}=0.6250$ (light blue dot in Figure 2), and $\mathrm{preference}-{\mathrm{driven}}_{\left[0.9,0.8\right]}=0.8000$ (bright orange dot in Figure 2).

Next, we present different solution spaces obtained for successive confusion matrices. Figure 3 presents such solution spaces for the proposed measure for cm4 and cm8 confusion matrices (please refer to Figure 1, which can be described as an opposition to each other, and thus they were selected for the analysis. The value of $\kappa_{1}$ increases, and the value of $\kappa_{2}$ decreases (the value of the proposed measure for cm4 decreases, while that for cm8 increases—and vice versa).

Different solution spaces for the confusion matrices from Figure 1 are presented in Figure 4 and Figure 5. In the first case (Figure 4), specific confusion matrices are selected:cm1, which represents the ideal classification—in Figure 4a, it can be seen that the value of the $\mathrm{preference}-{\mathrm{driven}}_{\vec{\kappa }}$ measure is the same for each $\vec{\kappa }$ vector;cm9, which represents a classification in which all samples are assigned to only one class (in this case, the first one)—in Figure 4b, it can be seen that $\kappa_{2}$ does not affect the value of the $\mathrm{preference}-{\mathrm{driven}}_{\vec{\kappa }}$ measure;cm4 and cm8, which were previously presented in Figure 3, but this time are shown separately in Figure 4c,d, respectively—they represent an example, not an extreme, case of classification.

Figure 5 presents the solution spaces for all (except cm9) confusion matrices from Figure 1. It allowed us to present the dynamics of the solution space based on the $\mathrm{preference}-{\mathrm{driven}}_{\vec{\kappa }}$, depending on the classification being evaluated and the value of the $\vec{\kappa }$ vector. Depending on the values of the $\vec{\kappa }$ vector, the proposed measure differently evaluates the classifier. It is also presented in Table 3, where the results for different values of the $\vec{\kappa }$ vector are presented. Examples [0.3,0.6], [0.9,0.4], and [0.5,0.5] have been chosen, as well as precision (which is equivalent to $\vec{\kappa }=\left[0.0,0.0\right]$) and recall (equivalent to $\vec{\kappa }=\left[1.0,1.0\right]$). As can be seen, depending on the given preferences, the evaluation of the classifier even in this case changes noticeably.

Analogously to Table 3, Figure 6 presents the classification quality assessment values for each of the prepared confusion matrices. Such a visualization allows us to observe the influence of the proposed measure on the classification evaluation (compare with the earlier discussed example for cm4 and cm8 for which precision, recall, and F1 have identical values), where diagonal lines allow a more straightforward analysis of changes between sample confusion matrices; in this case, we can see that the mentioned measures have identical values for these confusion matrices, but the proposed preference-driven measure obtains different scores for the same preference vectors. Please note that confusion matrices should not be interpreted as consecutive occurrences. Classification evaluation values should be compared regardless of the order in which they are presented.

To further compare the proposed preference-driven measure with the F1 measure, which is also based on recall and precision values, we analyzed the values of the preference vector ($\vec{\kappa }$) that produce the same classification score as the F1.

We performed careful analyses for all the confusion matrices described in this section, except for cm1, for which the value of all measures is always 1.0000 (this confusion matrix represents the ideal classification). Our observations indicate that there are (unlike the recall and precision measures) no classical values for the preference vector to find the equivalent value of the F1 measure. Therefore, for subsequent confusion matrices, the F1 counterparts are, respectively, preference-driven measures with preference vectors: for cm2, it is $\vec{\kappa }=\left[0.015,0.095\right]$, then for cm3, it is $\vec{\kappa }=\left[0.010,0.145\right]$; for cm4, it is $\vec{\kappa }=\left[0.189,0.284\right]$; for cm5, it is $\vec{\kappa }=\left[0.045,0.095\right]$; for cm6, it is $\vec{\kappa }=\left[0.06,0.12\right]$; for cm7, it is $\vec{\kappa }=\left[0.5,0.5\right]$; for cm8, it is $\vec{\kappa }=\left[0.284,0.189\right]$, and for cm9, it is $\vec{\kappa }=\left[0.67,0.00\right]$.

These observations indicate significant differences between the F1 and the proposed preference-driven measures. As we have already presented, F1 for a single confusion matrix (i.e., the classifier score) is always represented by a single value. In contrast, it is possible to control the quality score according to preferences in the preference-driven case.

### 4.3. Default Values of the Preference Vector

The proposed measure is used to evaluate the classifier’s quality as closely as possible to the stated preferences (as long as the decision maker clearly describes these preferences). To make the measure more comparable with other measures, we propose, as an alternative, the default values of the preference vector for the $\mathrm{preference}-{\mathrm{driven}}_{\vec{\kappa }}$ measure.

The ratio of the number of objects in each class to the number of all samples is taken as the default value of the preference vector. This means that, for classes with many samples, higher weight is related to precision for this class, while, for classes with a relatively small number of samples, higher weight is related to recall of this class. Such a solution allows compensation for the situation in which the samples are more often classified into classes with many samples. When validating a classifier, there is always a learning set (whether for train-and-test or cross-validation), which each time allows the mentioned default values of the preference vector to be determined—the preference vector can correspond to the distribution of cases in the learning data.

In analogy with previously adopted designations, a notation for the default value of the $\vec{\kappa }$ vector in Formula (Equation 31) is proposed. The designations are the same as those contained in Table 2; additionally, *s* is the sum of all samples.
(31)$$\vec{\kappa }=\left[\frac{TP_{1}+FN_{1}}{s},\frac{TP_{2}+FN_{2}}{s},\cdots ,\frac{TP_{i}+FN_{i}}{s},\cdots ,\frac{TP_{C-1}+FN_{C-1}}{s},\frac{TP_{C}+FN_{C}}{s}\right]$$

For example, in the situation described in Section 4.2, where the classes in the confusion matrix are at equilibrium, the implicit vector would be of the form $\vec{\kappa }=\left[0.5,0.5\right]$. Its value is incidentally given in Table 3, in column $\mathrm{preference}-{\mathrm{driven}}_{\left[0.5,0.5\right]}$.

Extending the example, a confusion matrix is proposed, shown in Table 4. In this case, there are three classes, containing 50, 20, and 30 samples each, respectively. Therefore, the default values of the preference vector would be as follows:$$\vec{\kappa }=\left[\frac{50}{100},\frac{20}{100},\frac{30}{100}\right],$$
so
$$\vec{\kappa }=\left[0.5,0.2,0.3\right].$$
Here, 50 out of 100 instances are in class 1, 20 out of 100 instances are in class 2, and 30 out of 100 instances are in class 3. The preference-driven measure value is 0.656, because
$$\begin{array}{ccc} \mathrm{preference}-{\mathrm{driven}}_{\left[0.5,0.2,0.3\right]} & = & \frac{1}{3}\times \left(0.5\times \frac{TP_{1}}{TP_{1}+FP_{1}}+\left(1-0.5\right)\times \frac{TP_{1}}{TP_{1}+FN_{1}}\right)+\\ & & \frac{1}{3}\times \left(0.2\times \frac{TP_{2}}{TP_{2}+FP_{2}}+\left(1-0.2\right)\times \frac{TP_{2}}{TP_{2}+FN_{2}}\right)+\\ & & \frac{1}{3}\times \left(0.3\times \frac{TP_{3}}{TP_{3}+FP_{3}}+\left(1-0.3\right)\times \frac{TP_{3}}{TP_{3}+FN_{3}}\right) \end{array}$$
so
$$\begin{array}{ccc} \mathrm{preference}-{\mathrm{driven}}_{\left[0.5,0.2,0.3\right]} & = & \frac{1}{3}\times \left(0.5\times \frac{40}{40+8+9}+0.5\times \frac{40}{40+7+3}\right)+\\ & & \frac{1}{3}\times \left(0.2\times \frac{10}{10+7+1}+0.8\times \frac{10}{10+8+2}\right)+\\ & & \frac{1}{3}\times \left(0.3\times \frac{20}{20+3+2}+0.7\times \frac{20}{20+9+1}\right) \end{array}$$
$$\mathrm{preference}-{\mathrm{driven}}_{\left[0.5,0.2,0.3\right]}=\frac{1}{3}\times \left(0.5\times 0.702+0.5\times 0.8+0.2\times 0.556+0.8\times 0.5+0.3\times 0.8+0.7\times 0.667\right)$$
$$\mathrm{preference}-{\mathrm{driven}}_{\left[0.5,0.2,0.3\right]}=\frac{1}{3}\times \left(0.351+0.4+0.111+0.4+0.24+0.467\right)$$
$$\mathrm{preference}-{\mathrm{driven}}_{\left[0.5,0.2,0.3\right]}=0.656$$

The results for the default values of the preference vector for the $\mathrm{preference}-{\mathrm{driven}}_{\vec{\kappa }}$ measure are also analyzed in Section 5, when testing the proposed measure with different real data sets.

## 5. Analysis on Real-World Data Sets

As the paper proposes a new classification quality assessment measure whose value depends on the stated preferences (called preference vector), we conducted experiments on real-world data sets. First, we checked the importance of the proposed measure depending on the given preference vector—while comparing the performance and ranks of the classifiers with classical measures of classification quality assessment.

### 5.1. Experiment Conditions

The proposed measure $\mathrm{preference}-{\mathrm{driven}}_{\vec{\kappa }}$ is compared with the classical measures described in this paper (see Section 3.2). As some of the measures, in the case of multi-class classification, reduce to the same measure, the following names are used in this section: accuracy (see Equation (Equation 16)), precision (see Equation (Equation 17)), recall (see Equation (Equation 18)), F1 (see Equation (Equation 23)), and MCC (see Equation (Equation 25)).

Four well-known data sets were selected for classification, whose structures are described in Table 5. As one can see, multi-class data sets were selected, in which, additionally, the distribution of samples in classes was uneven (see “percent of samples per class” in Table 5, where the ratio of samples of each class to the whole data set is given). On the other hand, as test classifiers, we selected classifiers available in the system Weka-3-6-11 [47]. More specifically, these were the following classifiers: Bagging [48], BayesNet [49], DecisionTable [50], C4.5 (J48) [51], and RandomForest [52]. In each case, 10-fold cross-validation was used so that each sample was subject to prediction.

Classification results from the Weka system (including confusion matrices) and the values of the proposed classification quality evaluation measure for fixed preference vectors are available on the website of the Department of Machine Learning of the University of Economics in Katowice (https://www.ue.katowice.pl/jednostki/katedry/wiik/katedra-uczenia-maszynowego/zrodla-danych/preference-driven.html (accessed on 6 February 2022)).

The number of decision classes in each data set is different, so the checked combinations of values in the preference vector determined different, possible values of subsequent elements of the vector. Thus, the number of combinations is close to 15,000 (except for the krkopt data set, which is too large and had to exceed this value). In this way, we obtained:car—[0, 0.1, 0.2, 0.3, 0.4, 0.5, 0.6, 0.7, 0.8, 0.9, 1], which gives 14,641 combinations (because: 4 decision classes and 11 values of κ, so $11^{4}=$ 14,641);nursery—[0, 0.166, 0.332, 0.498, 0.664, 0.83, 1], which gives $7^{5}=$ 16,807 combinations;dermatology—[0, 0.25, 0.5, 0.75, 1], which gives $5^{6}=$ 15,625; combinations;krkopt—[0.33, 0.66], which gives $2^{18}=$ 262,144 combinations.

### 5.2. Experimental Results

Table 6, Table 7, Table 8 and Table 9 present the results for all data sets and all classifiers. Ratings have been made for all measures, with $\mathrm{preference}-{\mathrm{driven}}_{\vec{\kappa }}$ determined each time for the default values and five different, selected preference vectors. Evaluation values were presented along with the ranking order of the classifier for each measure (in brackets). This approach allows us not only to evaluate the differences between the evaluation values but also to indicate whether, using the proposed measure, it could happen that another classifier would be better than those indicated by the classical classification quality evaluation measures. On the other hand, Figure 7 shows histograms of the $\mathrm{preference}-{\mathrm{driven}}_{\vec{\kappa }}$ measure values for all tested preference vectors.

Figure 7 presents 20 different histograms used to show the distribution of values (without any unnecessarily detailed information). In part (a), successive histograms are shown for the car data set, then (b) contains the nursery histograms, (c) is related to dermatology, and (d) is the krkopt results. This figure concerns the value of the preference-driven measure determined for different preference vectors ($\vec{\kappa }$). Therefore, the X-axis represents the value of the preference-driven measure, while the Y-axis represents the number of occurrences of this value.

This presentation in Figure 7 allows us to notice that in each case, using the proposed measure, it is possible to obtain a different score with the same confusion matrix. In addition, in most cases, the values of the measure are close to a normal distribution. A slightly more interesting case is (b), with the Bagging, BayesNet, and RandomForest classifiers. In this case, however, it should be noted that the classification each time gives a specific confusion matrix (detailed results are available on the UE Katowice website (https://www.ue.katowice.pl/jednostki/katedry/wiik/katedra-uczenia-maszynowego/zrodla-danych/preference-driven.html) (accessed on 6 February 2022)). Note that there are only two samples in the first class. The class with the most significant number of samples is always correctly identified. Additionally, in case (c), the particular histogram is obtained for the RandomForest classifier, where the classification was excellent, close to error-free. In contrast, the values were rounded to the second decimal for the histogram, hence such a high concentration of preference-driven measure values.

Analyzing the experimental results, it is worth noting that by using different vectors, in the case of a preference-driven measure, it is possible to indicate a different classifier as more adapted to the problem. For example, in the case of the car data set (see Table 6), the classifier DecisionTable turns out to be the best for some preference vectors (e.g., [0,1,1,0.9]); similarly, the classifier BayesNet turns out to be better than Bagging if the preference vector is, among others, [0.3,1,1,1]. The situation is similar for the set of nursery data (see Table 7), where also DecisionTable turns out to be the best, while the preference vector is [0.498,1,0.166,1,0]. A similar difference, but on subsequent ranks (between 3 and 4), can be observed also in the case of the krkopt data set (see Table 9). Although, in the case of the dermatology data set (see Table 8), no change of ranks was observed, it could be observed that the disproportion between the assessment of specific classifiers changed a lot and, e.g., in the case of vector [0,0,0,1,1,0.75], the difference between the best BayesNet and the second RandomForest was less than 0.0083, where, with classical measures of classification quality assessment, this difference was around 0.02.

The aim of the experiment was to check whether the proposed measure will show different classifiers for the same problem depending on the value in the preference vector. It turned out that, indeed, the measure indicates different classifiers, even with a limited number of vectors tested.

## 6. Discussion

In supervised learning, one of the necessary steps in adequately designed research is teaching on training data different classifiers, sometimes with different parameters, and then evaluating their quality. In our work, we assume that the user has specific preferences. In addition, they relate to particular decision classes. Thus, we can immediately identify a suitable classifier.

Using a vector of preferences, it is possible to indicate whether precision or recall is more important. It is crucial for many problems to indicate that it is possible to select a classifier better adapted to the structure of the data and decision makers’ expectations for each class separately. The best strategy is to examine the training of as many classifiers as possible (with different parameters). Such an approach allows the identification of the potentially best classifiers for a given problem.

The proposed preference-driven measure used in the experiments allowed for a better selection of the classifier most suitable for the task. With an approach allowing us to indicate whether precision or recall is more important—for each class separately—it is possible to select a classifier more adapted to the more important (from the evaluation point of view) decision class.

Despite the limited number of combinations, the conducted experiments indicated that the proposed measure, depending on the preferences conveyed in the form of a preference vector, points to different classifiers as the best choice for further prediction.

Since the set of values in the preference vector is infinite, the measurement values are also unlimited. Therefore, the calculation of a measure and comparison with other measures is possible only by calculating them at specific points. It should be noted that it is not necessary to test many combinations of the preference vector in a real application. Instead, these should be predetermined, indicating the relative importance of the recall–precision balance in each decision class.

This also distinguishes the preference-driven measure from the $F_{\beta }$ measure, which also raised attempts to weigh between precision and recall. However, the proposed preference-driven measure is different. Note that, in the case of $F_{\beta }$, there is a weighting between precision and recall overall for the classifier. In contrast, there are weights for each decision class with the preference-driven measure. It allows us to change the emphasis on precision and recall depending on each decision class (differently). In the case of $F_{\beta }$, this possibility does not exist.

## 7. Conclusions

This article presents a new idea for a preference-driven classification measure. We tried to show that the measure works, i.e.,
for different preference vectors, different classifiers are more advantageous then others;the obtained results of the comparison make it easier for the user to understand the effects of classification and make the right decision as to which classifier to use.

In the “objective” approach (without preferences), the result is unambiguous and comparable, but the best classifier will not necessarily be adjusted to the subjective needs of the user. In the “subjective” approach (preference-driven), comparability is difficult, but in return, users can acquire a classifier better suited to their requirements.

The concept of this measure results from the shortcomings of measures related to the multi-class classification. Nowadays, we observe a large number of classification methods. There is no single versatile classification measure capable of catching up on concepts related to both: overall good classification quality and a particular focus on the selected decision classes. The whole idea of different classification measures is mostly extended for the binary decision classes, which often fail to achieve good results for real-world data. At the same time, multi-class classification measures are based on averaging the results, which can be fair for general cases but fails to include decision makers’ preferences related to the particular classes.

Similarly, for unbalanced cases, where there is a need to focus on particular classes, the proposed preference-driven measure fits well for this gap. To be more precise, our proposed preference-driven measure can be aligned with the decision makers’ preferences regarding the relative importance of precision and recall. We also present the idea of setting the default values for the vector of preferences based on the overall number of samples assigned to every decision class. The most important advantage of the proposed idea is the good fit between the well-known measures such as precision and recall. Moreover, the κ preference vector allows us to direct the focus to a particular decision class, or even to express the importance of selected decision classes in terms of the precision measure (and others in terms of recall).

At the same time, we show that even for potentially trivial cases, such preference-driven measures could lead to entirely different results based on the κ selection. It opens a discussion for multi-class classification and leads to an interesting situation. The solution for the classification problem should not be considered a single scalar value.

In the future, a preference-driven measure can be used in line with the proposed approach. Alternatively, the factors of which the measure is composed could be scrutinized, and other measures could be used instead of the class’s relative precision and recall values.

## Figures and Tables

**Figure 1 entropy-24-00531-f001:**
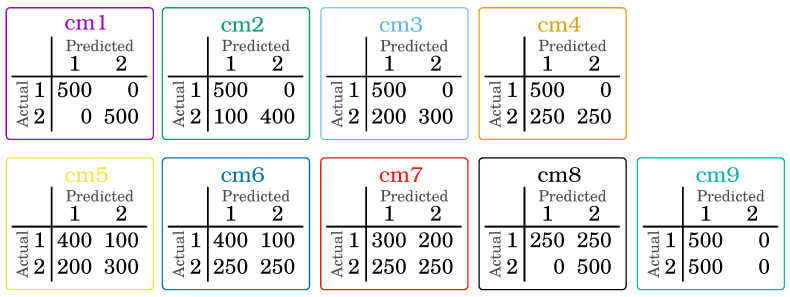
Confusion matrix used for measure analysis.

**Figure 2 entropy-24-00531-f002:**
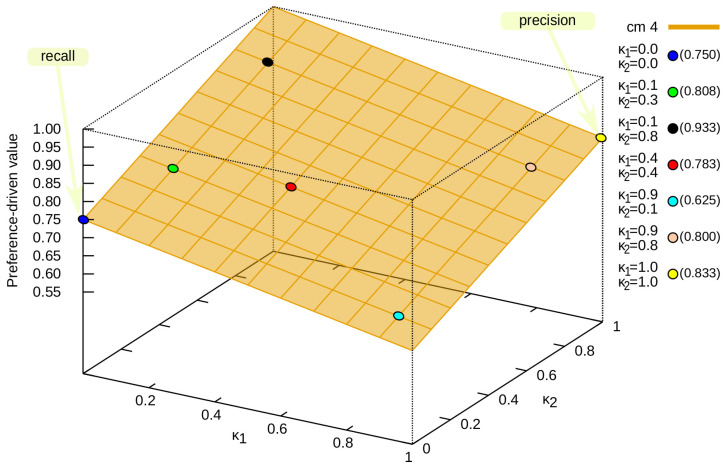
All possible values of the preference-driven measure for the confusion matrix cm4.

**Figure 3 entropy-24-00531-f003:**
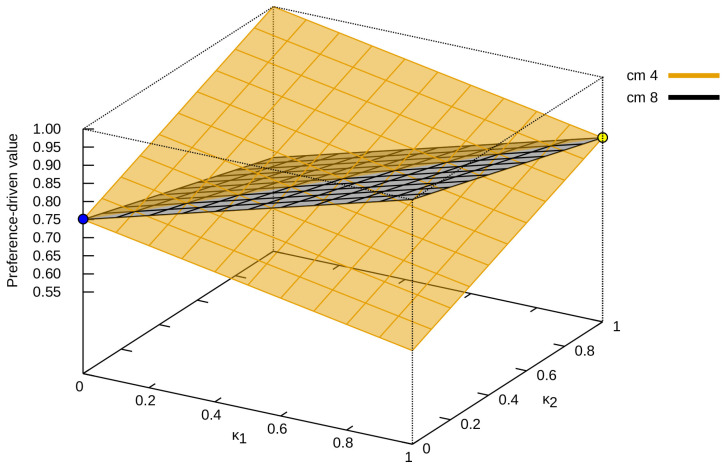
All possible values of the preference-driven measure for the confusion matrix cm4 and its opposite cm8.

**Figure 4 entropy-24-00531-f004:**
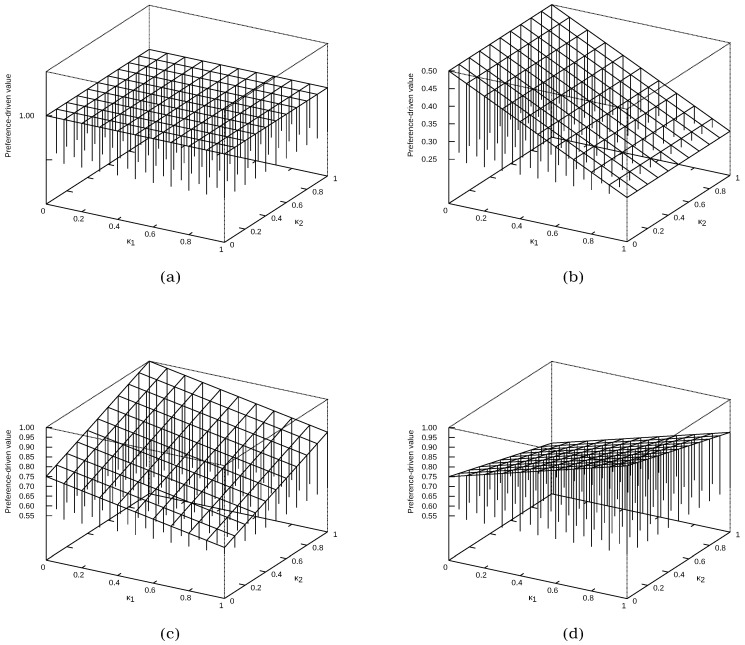
Example distributions of values of the preference-driven measure: (**a**) confusion matrix cm1; (**b**) confusion matrix cm9; (**c**) confusion matrix cm4; (**d**) confusion matrix cm8.

**Figure 5 entropy-24-00531-f005:**
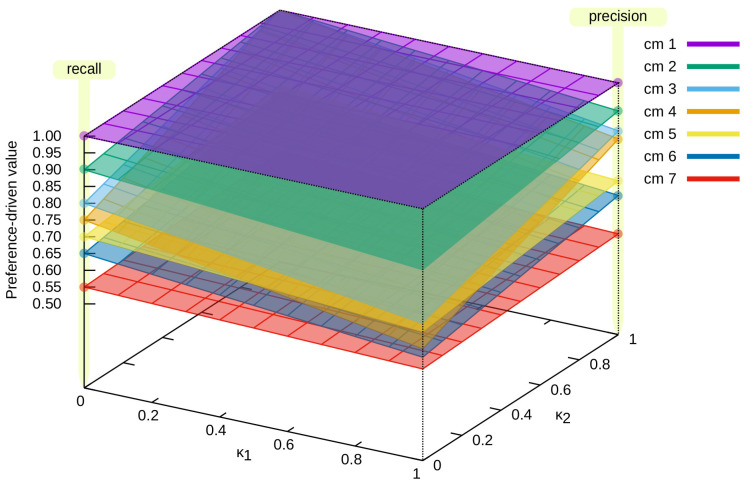
Solution spaces of proposed measure values for selected confusion matrices.

**Figure 6 entropy-24-00531-f006:**
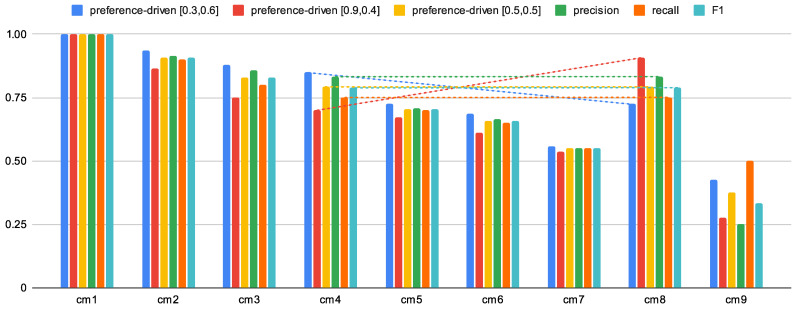
Example values of the proposed measure in comparison with other measures for assessing the quality of classification (the diagonal lines allow easier analysis of the changes between the sample confusion matrices).

**Figure 7 entropy-24-00531-f007:**
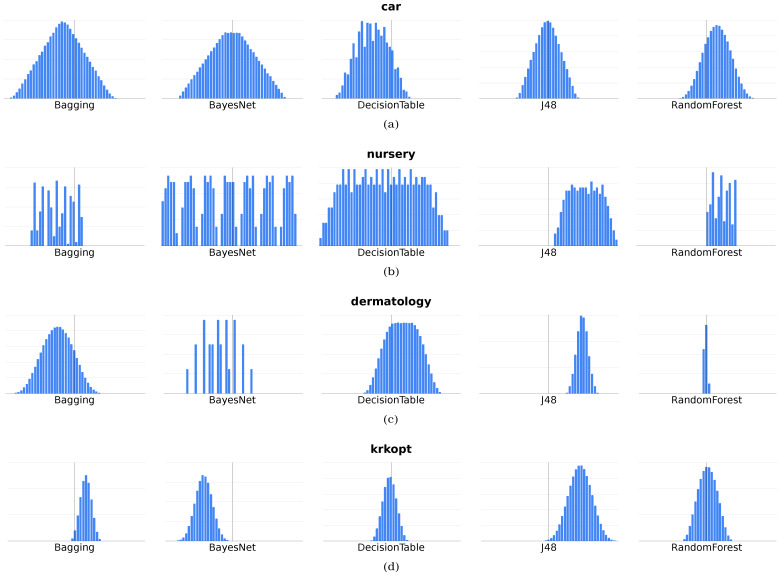
Histogram of preference-driven measure values in the (**a**) car data set; (**b**) nursery data set; (**c**) dermatology data set; (**d**) krkopt data set—the X-axis represents the value of the preference-driven measure, while the Y-axis represents the number of occurrences of this value.

**Table 1 entropy-24-00531-t001:** Confusion matrix for binary classification.

	Predicted Class 1	Predicted Class 2
**Actual class 1**	TP	FN
**Actual class 2**	FP	TN

**Table 2 entropy-24-00531-t002:** Confusion matrix for multiple classes.

	Predicted Class 1	Predicted Class 2	Predicted Class 3	⋯	Predicted Class i	⋯	Predicted Class C-1	Predicted Class C
		$FP_{2}$	$FP_{3}$		$FP_{i}$		$FP_{C-1}$	$FP_{C}$
**Actual class 1**	$TP_{1}\;TN\backslash_{\left\{1\right\}}$	$TN\backslash_{\left\{1,2\right\}}$	$TN\backslash_{\left\{1,3\right\}}$	⋯	$TN\backslash_{\left\{1,i\right\}}$	⋯	$TN\backslash_{\left\{1,C-1\right\}}$	$TN\backslash_{\left\{1,C\right\}}$
		$FN_{1}$	$FN_{1}$		$FN_{1}$		$FN_{1}$	$FN_{1}$
	$FP_{1}$		$FP_{3}$		$FP_{i}$		$FP_{C-1}$	$FP_{C}$
**Actual class 2**	$TN\backslash_{\left\{1,2\right\}}$	$TP_{2}\;TN\backslash_{\left\{2\right\}}$	$TN\backslash_{\left\{2,3\right\}}$	⋯	$TN\backslash_{\left\{2,i\right\}}$	⋯	$TN\backslash_{\left\{2,C-1\right\}}$	$TN\backslash_{\left\{2,C\right\}}$
	$FN_{2}$		$FN_{2}$		$FN_{2}$		$FN_{2}$	$FN_{2}$
	$FP_{1}$	$FP_{2}$			$FP_{i}$		$FP_{C-1}$	$FP_{C}$
**Actual class 3**	$TN\backslash_{\left\{1,3\right\}}$	$TN\backslash_{\left\{2,3\right\}}$	$TP_{3}\;TN\backslash_{\left\{3\right\}}$	⋯	$TN\backslash_{\left\{3,i\right\}}$	⋯	$TN\backslash_{\left\{3,C-1\right\}}$	$TN\backslash_{\left\{3,C\right\}}$
	$FN_{3}$	$FN_{3}$			$FN_{3}$		$FN_{3}$	$FN_{3}$
**⋯**	⋯	⋯	⋯	⋯	⋯	⋯	⋯	
	$FP_{1}$	$FP_{2}$	$FP_{3}$				$FP_{C-1}$	$FP_{C}$
**Actual class i**	$TN\backslash_{\left\{1,i\right\}}$	$TN\backslash_{\left\{2,i\right\}}$	$TN\backslash_{\left\{3,i\right\}}$	⋯	$\;TP_{i}\;\;TN\backslash_{\left\{i\right\}}$	⋯	$TN\backslash_{\left\{i,C-1\right\}}$	$TN\backslash_{\left\{i,C\right\}}$
	$FN_{i}$	$FN_{i}$	$FN_{i}$				$FN_{i}$	$FN_{i}$
⋯	⋯	⋯	⋯	⋯	⋯	⋯	⋯	⋯
	$FP_{1}$	$FP_{2}$	$FP_{3}$		$FP_{i}$			$FP_{C}$
**Actual class C-1**	$TN\backslash_{\left\{1,C-1\right\}}$	$TN\backslash_{\left\{2,C-1\right\}}$	$TN\backslash_{\left\{3,C-1\right\}}$	⋯	$TN\backslash_{\left\{i,C-1\right\}}$	⋯	$TP_{C-1}\;TN\backslash_{\left\{C-1\right\}}$	$TN\backslash_{\left\{C,C-1\right\}}$
	$FN_{C-1}$	$FN_{C-1}$	$FN_{C-1}$		$FN_{C-1}$			$FN_{C-1}$
	$FP_{1}$	$FP_{2}$	$FP_{3}$		$FP_{i}$		$FP_{C-1}$	
**Actual class C**	$TN\backslash_{\left\{1,C\right\}}$	$TN\backslash_{\left\{2,C\right\}}$	$TN\backslash_{\left\{3,C\right\}}$	⋯	$TN\backslash_{\left\{i,C\right\}}$	⋯	$TN\backslash_{\left\{C-1,C\right\}}$	$TP_{C}\;TN\backslash_{\left\{C\right\}}$
	$FN_{C}$	$FN_{C}$	$FN_{C}$		$FN_{C}$		$FN_{C}$	

**Table 3 entropy-24-00531-t003:** Example values of the proposed preference-driven measure in comparison with other measures for assessing the quality of classification (p-d is the abbreviation for the preference-driven measure). Results determined for all confusion matrices presented in Figure 1.

	${p-d}_{\left[0.3,0.6\right]}$	${p-d}_{\left[0.9,0.4\right]}$	${p-d}_{\left[0.5,0.5\right]}$	Precision ^1^	Recall ^2^	F1 ^3^
cm1 ^4^	1.0000	1.0000	1.0000	1.0000	1.0000	1.0000
cm2 ^4^	0.9350	0.8650	0.9083	0.9157	0.9000	0.9083
cm3 ^4^	0.8771	0.7514	0.8286	0.8571	0.8000	0.8276
cm4 ^4^	0.8500	0.7000	0.7917	0.8333	0.7500	0.7895
cm5 ^4^	0.7250	0.6700	0.7042	0.7083	0.7000	0.7041
cm6 ^4^	0.6866	0.6098	0.6574	0.6648	0.6500	0.6573
cm7 ^4^	0.5585	0.5366	0.5503	0.5505	0.5500	0.5502
cm8 ^4^	0.7250	0.9083	0.7917	0.8333	0.7500	0.7895
cm9 ^4^	0.4250	0.2750	0.3750	0.2500	0.5000	0.3333

^1^macro_precision (Equation (Equation 17)) is equal to $\mathrm{preference}-{\mathrm{driven}}_{\left[1.0,1.0\right]}$. ^2^
macro_recall (Equation (Equation 18)) is equal to $\mathrm{preference}-{\mathrm{driven}}_{\left[0.0,0.0\right]}$. ^3^
macro_F1 (Equation (Equation 23)). ^4^ See Figure 1.

**Table 4 entropy-24-00531-t004:** Example confusion matrix used to demonstrate how to determine the proposed preference-driven measure.

	Predicted Class 1	Predicted Class 2	Predicted Class 3
**Actual class 1**	40	7	3
**Actual class 2**	8	10	2
**Actual class 3**	9	1	20

**Table 5 entropy-24-00531-t005:** Characteristics of the real data sets used to test the proposed preference-driven measure and comparison with classical measures.

Data Set	Number of Samples	Number of Attributes	Number of Classes	Percent of Samples per Class
car	1728	6	4	0.70 0.22 0.04 0.04
nursery	12960	8	5	0.00 0.33 0.33 0.03 0.31
dermatology	366	34	6	0.17 0.31 0.20 0.14 0.13 0.05
krkopt	28056	6	18	0.10 0.00 0.00 0.01 0.00 0.01 0.02 0.02 0.02 0.05 0.06 0.07 0.10 0.13 0.15 0.16 0.08 0.01

**Table 6 entropy-24-00531-t006:** Results for the car data set—the value of the classification quality assessment (in brackets, we give the ranking of the classifier, according to the given measure). The ranking determines the order of the classifier depending on the classification quality rating measure used.

	Bagging	BayesNet	DecisionTable	C4.5 (J48)	RandomForest
accuracy	0.9167 (3)	0.8571 (5)	0.9149 (4)	0.9236 (2)	0.9462 (1)
precision	0.7586 (5)	0.7940 (4)	0.8557 (2)	0.8179 (3)	0.8565 (1)
recall	0.7665 (4)	0.6040 (5)	0.8177 (3)	0.8289 (2)	0.8581 (1)
F1	0.7625 (4)	0.6861 (5)	0.8363 (2)	0.8233 (3)	0.8573 (1)
MCC	0.8208 (3)	0.6737 (5)	0.8101 (4)	0.8345 (2)	0.8842 (1)
preference-driven ^1^	0.7670 (4)	0.6031 (5)	0.8171 (3)	0.8291 (2)	0.8584 (1)
$p-d_{\left[0,0.8,0.6,1\right]}$	0.7757 (4)	0.7582 (5)	0.8600 (1)	0.8295 (3)	0.8600 (1)
$p-d_{\left[0,1,0,1\right]}$	0.8021 (4)	0.6853 (5)	0.8631 (2)	0.8475 (3)	0.8647 (1)
$p-d_{\left[0,1,1,0.9\right]}$	0.7494 (5)	0.7954 (4)	0.8601 (1)	0.8127 (3)	0.8499 (2)
$p-d_{\left[0.3,1,1,1\right]}$	0.7555 (5)	0.8012 (4)	0.8609 (1)	0.8160 (3)	0.8534 (2)
$p-d_{\left[0.6,1,0.8,1\right]}$	0.7664 (5)	0.7743 (4)	0.8587 (1)	0.8233 (3)	0.8572 (2)

^1^ Default value of the preference vector (see Section 4.3).

**Table 7 entropy-24-00531-t007:** Results for the nursery data set—the value of the classification quality assessment (in brackets, we give the ranking of the classifier, according to the given measure). The ranking determines the order of the classifier depending on the classification quality rating measure used.

	Bagging	BayesNet	DecisionTable	C4.5 (J48)	RandomForest
accuracy	0.9737 (2)	0.9033 (5)	0.9470 (4)	0.9705 (3)	0.9909 (1)
precision	0.7518 (3)	0.7250 (5)	0.7661 (2)	0.7453 (4)	0.7849 (1)
recall	0.7226 (3)	0.5666 (5)	0.6722 (4)	0.7313 (2)	0.7765 (1)
F1	0.7369 (3)	0.6361 (5)	0.7160 (4)	0.7382 (2)	0.7806 (1)
MCC	0.9614 (2)	0.8579 (5)	0.9234 (4)	0.9568 (3)	0.9867 (1)
preference-driven ^1^	0.7224 (3)	0.5671 (5)	0.6727 (4)	0.7312 (2)	0.7764 (1)
$p-d_{\left[0,0,0,1,0\right]}$	0.7550 (3)	0.7362 (5)	0.7733 (2)	0.7470 (4)	0.7857 (1)
$p-d_{\left[0,0,0,1,1\right]}$	0.7524 (2)	0.7412 (5)	0.7520 (3)	0.7428 (4)	0.7850 (1)
$p-d_{\left[0,0.83,0,1,0\right]}$	0.7545 (3)	0.7228 (5)	0.7850 (2)	0.7490 (4)	0.7856 (1)
$p-d_{\left[0.498,1,0.166,1,0\right]}$	0.7544 (3)	0.7200 (5)	0.7874 (1)	0.7495 (4)	0.7855 (2)
$p-d_{\left[0.83,0.332,0.166,0,1\right]}$	0.7198 (3)	0.5662 (5)	0.6555 (4)	0.7280 (2)	0.7757 (1)

^1^ Default value of the preference vector (see Section 4.3).

**Table 8 entropy-24-00531-t008:** Results for the dermatology data set—the value of the classification quality assessment (in brackets, we give the ranking of the classifier, according to the given measure). The ranking determines the order of the classifier depending on the classification quality rating measure used.

	Bagging	BayesNet	DecisionTable	C4.5 (J48)	RandomForest
accuracy	0.3798 (5)	0.9727 (1)	0.8388 (4)	0.9454 (3)	0.9536 (2)
precision	0.3798 (5)	0.9692 (1)	0.8549 (4)	0.9372 (3)	0.9492 (2)
recall	0.3947 (5)	0.9707 (1)	0.8204 (4)	0.9368 (3)	0.9477 (2)
F1	0.3871 (5)	0.9699 (1)	0.8373 (4)	0.9370 (3)	0.9484 (2)
MCC	0.2170 (5)	0.9660 (1)	0.7982 (4)	0.9316 (3)	0.9418 (2)
preference-driven ^1^	0.3923 (5)	0.9708 (1)	0.8217 (4)	0.9368 (3)	0.9477 (2)
$p-d_{\left[0,0,0,0,1,0\right]}$	0.4090 (5)	0.9589 (1)	0.8030 (4)	0.9400 (3)	0.9506 (2)
$p-d_{\left[0,0,0,1,1,0.75\right]}$	0.4014 (5)	0.9589 (1)	0.8430 (4)	0.9369 (3)	0.9506 (2)
$p-d_{\left[0,0,1,1,0,1\right]}$	0.3807 (5)	0.9707 (1)	0.8800 (4)	0.9315 (3)	0.9477 (2)
$p-d_{\left[0.5,0,0.25,0.5,0.5,1\right]}$	0.3823 (5)	0.9699 (1)	0.8575 (4)	0.9375 (3)	0.9492 (2)
$p-d_{\left[0.5,1,0.75,0.5,0.5,0\right]}$	0.3921 (5)	0.9699 (1)	0.8178 (4)	0.9364 (3)	0.9477 (2)

^1^ Default value of the preference vector (see Section 4.3).

**Table 9 entropy-24-00531-t009:** Results for the krkopt data set—the value of the classification quality assessment (in brackets, we give the ranking of the classifier, according to the given measure). The ranking determines the order of the classifier depending on the classification quality rating measure used.

	Bagging	BayesNet	DecisionTable	C4.5 (J48)	RandomForest
accuracy	0.5872 (2)	0.3607 (5)	0.4908 (4)	0.5658 (3)	0.7025 (1)
precision	0.5735 (3)	0.3579 (5)	0.5784 (2)	0.5547 (4)	0.7377 (1)
recall	0.5406 (2)	0.2982 (5)	0.5187 (3)	0.5178 (4)	0.6628 (1)
F1	0.5566 (2)	0.3253 (5)	0.5469 (3)	0.5356 (4)	0.6982 (1)
MCC	0.5377 (2)	0.2784 (5)	0.4300 (4)	0.5135 (3)	0.6669 (1)
preference-driven ^1^	0.5404 (2)	0.2967 (5)	0.5191 (3)	0.5177 (4)	0.6629 (1)
$\mathrm{preference}-{\mathrm{driven}}_{\vec{\kappa }}$ ^2^	0.5776 (3)	0.3717 (5)	0.5806 (2)	0.5576 (3)	0.7426 (1)
$\mathrm{preference}-{\mathrm{driven}}_{\vec{\kappa }}$ ^3^	0.5542 (2)	0.3176 (5)	0.5415 (3)	0.5288 (4)	0.6869 (1)
$\mathrm{preference}-{\mathrm{driven}}_{\vec{\kappa }}$ ^4^	0.5554 (2)	0.3262 (5)	0.5462 (3)	0.5396 (4)	0.6925 (1)
$\mathrm{preference}-{\mathrm{driven}}_{\vec{\kappa }}$ ^5^	0.5741 (2)	0.3467 (5)	0.5472 (4)	0.5485 (3)	0.7287 (1)
$\mathrm{preference}-{\mathrm{driven}}_{\vec{\kappa }}$ ^6^	0.5638 (2)	0.3461 (5)	0.5587 (3)	0.5410 (4)	0.7033 (1)

^1^ Default value of the preference vector (see Section 4.3). ^2^
$\vec{\kappa }$ = [0, 1, 1, 1, 1, 1, 0.9, 0.9, 1, 0.9, 0.9, 0.9, 0, 0, 0, 0, 0.9, 1]. ^3^
$\vec{\kappa }$ = [0.33, 0.66, 0.33, 0.33, 0.66, 0.33, 0.66, 0.66, 0.33, 0.66, 0.66, 0.33, 0.66, 0.66, 0.66, 0.66, 0.33, 0.33]. ^4^
$\vec{\kappa }$ = [0.66, 0.33, 0.33, 0.33, 0.33, 0.66, 0.66, 0.33, 0.66, 0.66, 0.66, 0.33, 0.33, 0.66, 0.33, 0.33, 0.33, 0.33]. ^5^
$\vec{\kappa }$ = [1, 0.1, 1, 1, 1, 1, 0.1, 0.1, 0, 0.1, 0.1, 0.1, 0, 0, 0, 0, 0.9, 1]. ^6^
$\vec{\kappa }$ = [1, 1, 0.9, 0.9, 0.8, 0.8, 0.7, 0.6, 0.5, 0.4, 0.3, 0.3, 0.2, 0.2, 0.1, 0.1, 0.0, 0.0].

## Data Availability

The classification results (including confusion matrices) obtained in the Weka program, as well as the values of the preference-driven measure for the analyzed preference vectors, can be found on the website of the Department of Machine Learning of the University of Economics in Katowice: https://www.ue.katowice.pl/jednostki/katedry/wiik/katedra-uczenia-maszynowego/zrodla-danych/preference-driven.html (accessed on 6 February 2022). The implementation of the proposed preference-driven measure was prepared in Python language. The source code of this implementation is publicly available via GitHub: https://github.com/jankozak/preference-driven (accessed on 6 February 2022).

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
