# Peer review of "Preference-Driven Classification Measure"

_entropy, 2022, doi:10.3390/e24040531_

Round 1

Reviewer 1 Report

The paper proposes a classification performance measure which is a weighted average of precision and recall. These weights establish the relative importante of each class.

The topic of adequate measures for evaluating classification accuracy has merit and in fact each application domain tends to give preference to one measure over another. But the proposal does not seem original, albeit the plots of the variations of the performance measure according to different weights, which are not common in the related literature.

The F_beta-measure is originally an harmonic average of precision and recall, which is weighted by beta. In F1 the same weight is given for precision and recall, but this can be easily changed. Therefore, the only difference between the metric given in Equation 27 and taking the macro F-beta measure would be using a standard weighted average instead of the harmonic mean. But the idea of the harmonic mean is to  penalize better low precision/recall values. Therefore, why would it be better to withdraw the harmonic average? There is no discussion about that in the text and the relation to the well known F-beta measure is also absent.

In addition, it is usually very difficult to define proper costs in cost-sensitive metrics. Using the proportion of examples per class seems a natural choice, but is not new either. I did not understand, for  instance, what do you mean by "there is always a test set"in page 12, line 441. I did not understand how the ranges of values were defined in lines 486 to 488 either and the resulting number of combinations, maybe you could add some more text to clarify that.

Other remarks on the text are:

  • The phrase "we still lack a single and effective measure of classification quality independent of the data" is misleading. In fact, you cannot measure classification quality independently of any data. I believe what you wanted to express is that you want a general measure of quality independent on specific characteristics of the dataset, such as number of classes.
  • Page 2, lines 68 to 73: 1 section should be Section 1, 2 section should be Section 2, 4 section should be Section 4 and 5 section should be Section 5.
  • Page 3, line 98: the authors who? I believe they are Gösgens et al. (2021). 
  • Page 3, lines 107 to 110: you mention some standard threshold-based performance metrics and say they can be helpful for balanced or imbalanced datasets. But some of the measures listed, such as accuracy, as misleading for imbalanced classification problems. Please rephrase accordingly.
  • Page 4, line 189: I believe you must have A_1 x A_2 x ... x A_m.
  • Page 5, line 213: please remove the terms "both".
  • Page 7, line 272: "has" should be "have".
  • Page 7, lines 284 onwards: you were using n for number of observations in the dataset. According to Equation 3, C should be the number of classes. Please standardize the notations.
  • Page 7: are equations 13 and 14 correct? I do not understand how they are equivalent.
  • Page 8: the notation in Table 2 is misleading. 
  • Page 10, line 380: please remove the point after recall.
  • Page 10, line 302: the comma should be in the previous line.
  • Figure 5: what are the diagonal lines in this plot showing? Please state that in the caption of the figure.
  • References 38 and 42 are incomplete.

Author Response

Dear Reviewer,

We wish to thank you for your careful reading of the manuscript and for a set of helpful comments. We hope that after the revision the manuscript will find the approval of the editorial board. Here is the list of all changes we made in reaction to your suggestions.

With kindest regards,
Authors of the paper

Reviewer 2 Report

The manuscript addresses the research question on how to evaluate a classification algorithm, especially when dealing with multi class classification. The authors solve this question by defining and proposing a new measure based on the analyst preferences and the combination of the state of the art precision and recall measures. The new measures is similar to the well known F-measure.

They show the validity of their measure by applying it on a binary  and a multi-class classification problem obtaining promising results.

The proposed measure seems to be sound and the manuscript is well structured. Although I request following revisions before publication.

General revisions:

  • English language needs improvements, I suggest a native speaker
  • A better comparison of the authors’ measure with the F-measure is needed
  • The authors’ measure should be given a name
  • Tables 1, 2, 4, 5 need a better graphical design
  • Captions of tables need improvements
  • Conclusions are missing in the abstract
  • The authors publish results of their performed experiments and the values of the proposed classification quality evaluation measure on their website. The experimentation was performed by using Weka.
  • The authors should also publish an implementation (source code) of their measure and possibly integrate it in weka

Author Response

(The authors gave the same response as above.)

Reviewer 3 Report

This paper presents a weighted average measure for multiclass classification problems.

But the fundamentals behind the hyphotesis of the paper are questionable. Authors propose basically a weighted measure, where the weights come from a "preference" vector. Beyond the relationship between the size of the data sample and the weights, the rest is based on user's preference. There is no need to obtain a subjective measurement. If users are more interested in one class or Type I or II errors, it is straightforward to them to adjust the classifier (or detector) parameters to optimize that goal. 

In fact this is how real world detectors or classifiers are trained and deployed (most of them have a false alarm rate and then de detector is optimized). I understand other approaches depending on the cost of errors, but that is the point: the paper would be much more interesting if it follows a Bayesian approach and include some cost and loss functions where it is clear how the priors are affecting the final measure. An then provide some probabilistic well grounded guidance for users to define the "unique" quality measure number.

There is not a unified measure (beyond F measure and averages of each class quality measurements) simply because it is not possible. Any simplification to reduce all the information to a single number is just that, a simplification. It is more important that the user understands the results and take the corresponding decissions in the design of the classifier than providing to him with a number as if that number where the golden solution. To ontain false conclusions from some results is a very common mistake. I appreciate the efforts by authors to try to unify all of these, but it is simply impossible.

Author Response

(The authors gave the same response as above.)

Reviewer 4 Report

The paper presents a measure for supervised classification. The paper is well written, the aim and the motivation of the work are presented clearly.

The proposed measure is dedicated primarily to imbalanced datasets. It enables to set a preference for a particular decision class. Although authors propose an algorithm for setting default preferences, users can set it according to their needs. The idea is interesting and original. However, as the trained classification model operates on new, unknown data, I’m wondering if the optimal setting of the preference vector is possible. Maybe the authors could extend the discussion section in this area.

Author Response

(The authors gave the same response as above.)

Round 2

Reviewer 1 Report

I appreciate the effort the authors have made to improve the first version of the paper. But I feel there are still some corrections needed, as discussed next.

When you say in lines 128 to 130 that the extension of measures to the multiclass scenario in the literature is done by averaging the results achieved for the individual classes, it seems your measure works differently. But in fact Equation 30 is also an average of the precision and recall for the individual classes, but weighted. Maybe some rephrasing is necessary. 

The way F1 and F_beta (and their corresponding macro/micro versions) are presented gives the impression they are different measures. But F1 is an instantiation of F_beta using beta = 1. In fact, the most common instantiation of the F_beta measure is F1 and not F2 or F0.5 as stated in the text. Please rephrase accordingly.

The notation is still inconsistent. For Equations 14 to 30, when n is present, it should be replaced by C, the number of classes.

Some other minor reviews required:

  • Line 22: "of a general" should be "a general";
  • Line 149: "In" should be "in";
  • Line 302: the comma after 1 should be removed;
  • Line 311: <> in the ranges should be [] and the comma should be removed.
  • Does the phrase between parenthesis in lines 500 to 504 should be so? I think it is part of the main text and the parenthesis should be removed, please verify.
  • Line 529: the ":" symbol should be removed;
  • Line 535: I did not understand the term "this each", maybe the "this" should be removed?
  • Line 567: "structure is" should be "structures are"; 
  • Line 574: how many folds are used in your cross-validation?
  • Table 5: "Ratio of the share of decision classes" can be "% os samples per class";
  • What are the x and y axes in Figure 7? I do not understand some negative results in the x axis, for instance.

Author Response

(The authors gave the same response as above.)

Reviewer 2 Report

The authors performed my

requested revisions and

the manuscript has been 

improved.

I suggest acceptance

Author Response

Dear Reviewer,

Thank you very much for your contribution. It made our article even better

With kindest regards,
Authors of the paper

Reviewer 3 Report

Authors answer and discuss my comments in a solid way. Although we have philosophical differences about how to introduce priors in the optimization, the paper is well writen.

Author Response

(The authors gave the same response as above.)
